# Global tropical dry forest extent and cover: A comparative study of bioclimatic definitions using two climatic data sets

Jonathan Pando Ocón[1]*, Thomas Ibanez[2], Janet Franklin[3‡], Stephanie Pau[4‡], Gunnar Keppel[5‡], Gonzalo Rivas-Torres[6,7,8], Michael Edward Shin[1], Thomas Welch Gillespie[1]

1 Department of Geography, University of California Los Angeles, Los Angeles, CA, United States of America, 2 AMAP, CIRAD, CNRS, INRAE, IRD, Univ Montpellier, Montpellier, France, 3 Department of Botany and Plant Sciences, University of California Riverside, Riverside, CA, United States of America, 4 Department of Geography, Florida State University, Tallahassee, FL, United States of America, 5 UniSA STEM and Future Industries Institute, University of South Australia, Adelaide, Australia, 6 Colegio de Ciencias Biológicas y Ambientales, Universidad San Francisco de Quito, Quito, Ecuador, 7 Wildlife Ecology and Conservation, University of Florida, Gainesville, FL, United States of America, 8 Instituto de Geografía, Universidad San Francisco de Quito, Quito, Ecuador

⊘ These authors contributed equally to this work.
‡ These authors also contributed equally to this work.
* jonocon@g.ucla.edu

**Data Availability Statement:** All relevant data are within the manuscript and its Supporting Information files.

## Abstract

There is a debate concerning the definition and extent of tropical dry forest biome and vegetation type at a global spatial scale. We identify the potential extent of the tropical dry forest biome based on bioclimatic definitions and climatic data sets to improve global estimates of distribution, cover, and change. We compared four bioclimatic definitions of the tropical dry forest biome–Murphy and Lugo, Food and Agriculture Organization (FAO), DryFlor, aridity index–using two climatic data sets: WorldClim and Climatologies at High-resolution for the Earth's Land Surface Areas (CHELSA). We then compared each of the eight unique combinations of bioclimatic definitions and climatic data sets using 540 field plots identified as tropical dry forest from a literature search and evaluated the accuracy of World Wildlife Fund tropical and subtropical dry broadleaf forest ecoregions. We used the definition and climate data that most closely matched field data to calculate forest cover in 2000 and change from 2001 to 2020. Globally, there was low agreement (< 58%) between bioclimatic definitions and WWF ecoregions and only 40% of field plots fell within these ecoregions. FAO using CHELSA had the highest agreement with field plots (81%) and was not correlated with the biome extent. Using the FAO definition with CHELSA climatic data set, we estimate 4,931,414 km$^2$ of closed canopy ($\geq$ 40% forest cover) tropical dry forest in 2000 and 4,369,695 km$^2$ in 2020 with a gross loss of 561,719 km$^2$ (11.4%) from 2001 to 2020. Tropical dry forest biome extent varies significantly based on bioclimatic definition used, with nearly half of all tropical dry forest vegetation missed when using ecoregion boundaries alone, especially in Africa. Using site-specific field validation, we find that the FAO definition using CHELSA provides an accurate, standard, and repeatable way to assess tropical dry forest cover and change at a global scale.

**Funding:** JPO received the 2019 Geospatial @ UCLA Summer Fellowship which helped fund this research. URL:https://gis.ucla.edu/. The funders had no role in study design, data collection and analysis, decision to publish, or preparation of the manuscript.

**Competing interests:** The authors have declared that no competing interests exist.

## Introduction

Tropical dry forest has been estimated to comprise 42% of all tropical forests and is believed to be one of the world's most endangered biomes [1–3]. Tropical dry forests provide the ecosystem services needed to support millions of subsistence farmers in some of the world's poorest areas, and higher population densities are driving the demand for energy and land leading to higher tropical deforestation rates in dry forest than humid forest [4, 5]. Additionally, tropical dry forests harbor unique and diverse ecological communities and their deforestation contributes to the steady erosion of Earth's biodiversity [6–8]. To protect this critically endangered and valuable resource, we need reliable estimates on the extent of the tropical dry forest biome and understand the degree of uncertainty around those estimates.

Terrestrial biomes have long been associated with climatic range limits [9, 10]. These foundational early descriptions of global biome patterns are complemented by a growing understanding of the relationship between climate and vegetation form and function [11–14]. Recent advances in global climate data sets are improving our understanding of the extent and distribution of biomes. WorldClim, the most widely used climate dataset in biogeography, provides global, gridded climate data at up to 1 km resolution interpolated from a network of weather stations [15], while the Climatologies at High-resolution for the Earth's Land Surface Areas (CHELSA) offer 1 km resolution climate data based on a quasi-mechanistically statistical downscaling of global circulation models [16]. Furthermore, global data sets for estimated potential evapotranspiration (PET) can now map aridity indices at 1 km resolution [17, 18].

Bioclimatic definitions are useful for estimating biome extent and forest cover at different spatial scales and to predict the dominant vegetation without the influence of humans or disturbances such as fire [14, 19]. However, bioclimatic definitions cannot differentiate between vegetation types, such as savannas, shrublands, woodlands, and deciduous to evergreen forests, whose climatic limits overlap [14, 20–22]. Thus, climatic definitions allow delimiting the potential extent of biomes, but not detailed mapping of vegetation type boundaries.

There have also been significant advances in spaceborne remote sensing of forest cover [23–26]. Forest cover for the first two decades of the 21st century can be mapped globally based on forest cover change data sets that contain forest cover and percent forest cover at a 30 m resolution from Landsat imagery for 2000 to 2020 [27]. Such global forest cover products can be used to evaluate the extent and change of forest cover within the climatic boundaries of different biomes and forest types, providing crucial information for conservation and land use planning [28, 29].

There has been a growing number of efforts to map forest extent and forest cover change in tropical dry forest regions, especially to identify dry forest regions with a high conservation priority at a global spatial scale [4, 8, 23, 29, 30]. Most biogeographical studies and conservation evaluations of tropical dry forest biomes at a global spatial scale use World Wildlife Fund (WWF) terrestrial ecoregions to establish the potential extent of this biome [3, 23, 27, 31, 32]. Currently, the WWF has defined 53 tropical and subtropical dry broadleaf forest ecoregions.

Previous global analyses of tropical dry forest cover have, however, also shown that the tropical dry forest vegetation type (e.g. canopy height $\geq$ 3 m, closed canopy $\geq$ 40%, drought-adapted species) has been misidentified within 500 m pixels as tropical and subtropical grasslands, savanna and shrubland, woodland and scrub; and desert and xeric shrubland ecoregions [23]. Thus, more precise knowledge of the extent and location of tropical dry forest biome is needed in order to track forest cover change and better understand threats (deforestation, fire, climate change) and the conservation status (endangered species, old growth, protected areas) of this tropical forest biome.

The range boundaries of the tropical dry forest biome should coincide with climatic thresholds related to temperature, mean annual rainfall, seasonality, and water deficit [17, 20, 33].

However, there are several bioclimatic definitions of tropical dry forest that can be used to assess the global extent of this biome independently from WWF ecoregion data [1, 8, 17, 30, 34]. Widespread consensus on the bioclimatic parameters defining the tropical dry forest biome extent is yet to be established [35] and no studies compare the predicted global or regional extent of the forest biome based on different definitions of bioclimatic suitability [5].

There are a number of ways to validate and refine bioclimatic assessments of the tropical dry forest biome. Results can be compared to local vegetation maps [33, 36], high resolution (< 1 m) remote sensing imagery [17], and field inventories or georeferenced specimens [8, 33, 36]. Field plots, however, offer the most robust approach to validating bioclimatic models because they provide high resolution data on location, species composition, and structure over a standardize area [37]. Furthermore, field ecologists are often familiar with a site's natural history and have identified the vegetation and function as a tropical dry forest. Global studies of plant and forest diversity [38] and structure [31] now contain over 100 field plots from tropical dry forests, although certain regions may be underrepresented.

This study has three objectives related to the global extent of the tropical dry forest biome and vegetation type. First, we compare the extent of tropical dry forest predicted by four bioclimatic definitions (Murphy and Lugo, Food and Agriculture Organization, DryFlor, aridity index) for the biome using two different climatic datasets (WorldClim and CHELSA) and compare these to widely-used WWF ecoregions. We would expect that there should be high agreement (e.g. 70%) between individual bioclimatic definitions and WWF ecoregions regardless of climate datasets used. Second, we identify if WWF ecoregions or bioclimatic definitions have the highest agreement with field plots defined as tropical dry forest. We would expect that there is high agreement among WWF ecoregions, bioclimatic definitions, and field plots. Third, we use the best definition to calculate tropical dry forest vegetation cover and change from 2000 to 2020. Global tropical dry forest cover estimates should be similar to other published global estimates.

## Materials and methods

### Climatic definitions of tropical dry forest

We computed the bioclimate envelope (or the potential extent) of tropical dry forest located between 30˚N and 30˚S using four different definitions (Table 1). For all these definitions, we first subsetted frost-free areas (> 0˚C) using minimum temperature of coldest month with mean annual temperature > 17˚C [1]. We defined tropical dry forest biome according to the four bioclimatic definitions by subsetting areas of relevant climate variables according to the thresholds or ranges (Table 1), and then overlaid (intersected) those subsetted areas to delimit where all climatic criteria co-occur.

**Table 1. Overview of bioclimatic definitions (Murphy and Lugo, Food and Agriculture Organization of the United Nations [FAO], DryFlor, Aridity Index) of the tropical dry forest biome.**

| Bioclimatic Definition | Source | Annual Precipitation, Dry Season |
|---|---|---|
| Murphy and Lugo | Murphy and Lugo 1986 | 250–2000 mm, 4–7 months ≤100 mm |
| FAO | Sunderland et al. 2015 | 500–1500 mm, 5–8 months ≤100 mm |
| DryFlor | Banda et al. 2016 | ≤1800 mm, 3–6 months ≤100 mm |
| Aridity Index* | Bastin et al. 2017 | 0.2 < aridity index < 0.65 |

* Calculated by taking the ratio of mean annual precipitation to potential evapotranspiration.

## Climate data sets

We used two different climatologies to compute the potential extent of tropical dry forest biome, WorldClim [15, https://www.worldclim.org/] and CHELSA [16, http://chelsa-climate.org/]. Both climatologies have a spatial resolution of 30 arc seconds, or ~1 km, but differ in the way they were computed. We used the 2nd version of the WorldClim climatology, released in 2017. This climatology was derived from weather station measurements (1970–2000) interpolated using thin-plate splines with covariates including elevation, distance to the coast and three satellite-derived covariates: maximum and minimum land surface temperature as well as cloud cover, obtained with the MODIS satellite platform [15]. Trabucco and Zomer (2019) provided aridity index and PET computed using the WorldClim climatology [18, https://cgiarcsi.community/2019/01/24/global-aridity-index-and-potential-evapotranspiration-climate-database-v2/]. The second climatology, CHELSA, was derived from a quasi-mechanistically statistical downscaling of the ERA interim global circulation model with a GPCC bias correction [16]. This climatology is based on averaged climatic condition between 1979–2013. CHELSA incorporates topoclimate (e.g., orographic rainfall and wind fields), which is highly relevant for islands. We used the ENVIronmental Rasters for Ecological Modeling R-package [39] to compute the PET and aridity index using the CHELSA climatology.

## WWF ecoregions

The WWF ecoregions are polygon shapefiles that included 867 land units classified into 14 different ecoregions [40]. Each represents large units of land or water containing a geographically distinct assemblage of species, natural communities, and environmental conditions [40]. WWF has identified 53 ecoregions that fall within the Tropical and Subtropical Dry Broadleaf Forest category (Fig 1) (S1 Appendix).

## Field data sets

Biodiversity data, including vegetation plots, are valuable for addressing macro-ecological questions about community patterns and processes [41, 42], as well as global conservation problems in the global change era [43]. We tested the occurrence of the tropical dry forest biome based on bioclimatic definitions against 540 verified locations of tropical dry forest vegetation compiled from two primary sources (Fig 1). First, we searched Web of Science (v.5.32), Scopus, and Google Scholar databases for peer-reviewed articles published between January 1990 and September 2019. We queried titles, abstracts, and keywords for the following terms: tropical\*dry\* forest\* plots\*. We selected the peer-reviewed articles based on three criteria:

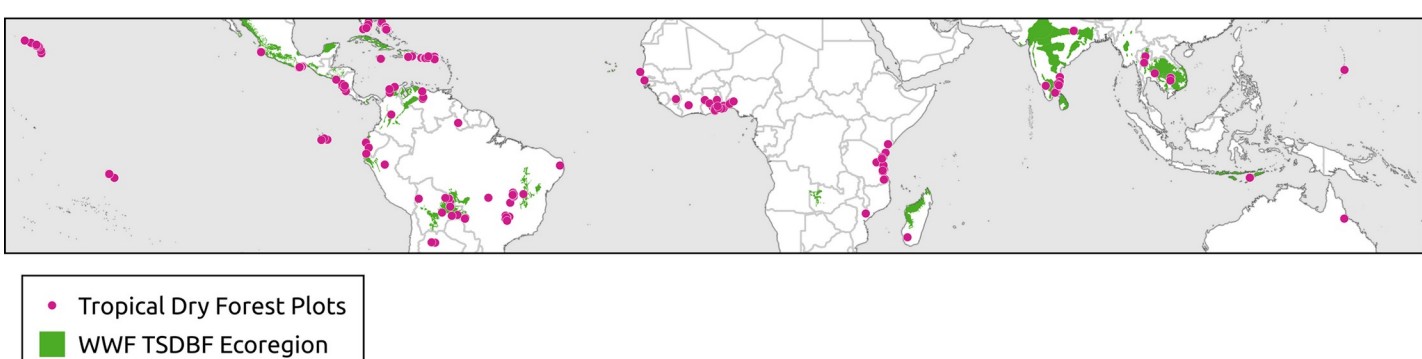

**Fig 1. Distribution of 540 tropical dry forest plots and the World Wildlife Fund's tropical and subtropical dry broadleaf forest ecoregion.**

1) plots classified as dry forest (closed-canopy) following author's classifications, and not savannas, woodlands (open-canopy), riparian or flooded forests, 2) articles needed to include published latitude and longitude of the plot location to within 1 km, and 3) forest had to be composed of drought tolerant tree species native to the region. Second, we searched four global forest data repositories for sites classified as tropical dry forest. Our search using the Global Biodiversity Information Facility, as well as ForestPlots.net, yielded much of the same plot data we had collected from our list of peer-reviewed studies. The Dryad data repository, however, yielded 40 additional dry forest plots from three peer-reviewed studies of dry forest across Latin America [44–46]. Additionally, we collected 150 sites from the United States Geological Survey's Forest Inventory and Analysis program covering tropical dry forest from the United States and territories (Hawai'i, Puerto Rico and US Virgin Islands). We used Google Earth to verify the coordinates were within areas with closed canopy forest and to establish seasonality using built-in, time-lapse imagery collected since 1984 across varying seasons. These plot locations were used to determine whether the tropical dry forest bioclimatic definitions accurately capture observed locations of the tropical dry forest vegetation type (S2 Appendix).

## Global forest cover and change

Originally developed to provide high-resolution global maps of forest cover change from 2000 to 2012 using Landsat 7 imagery, the Global Forest Change data set has grown to include time series analysis of Landsat 5, 7 and Landsat 8 imagery now covering forest cover and forest loss from 2000 to 2020 [18]. Each pixel has a spatial resolution of 1 arc second, or roughly 30 m. We calculate mean, minimum, and maximum percent canopy cover over each field plot at 1 km resolution to provide estimates of tropical dry forest cover and variability across the landscape in 2000 (S2 Appendix). Within the bioclimatic envelope of tropical dry forest biome that best matched the field plots, we identified forest areas as areas with a forest cover $\geq$ 40% [17, 23] for the year 2000, 2001 and 2020, and calculated forest cover change between 2001 and 2020.

## Data analyses

**Spatial coverage.**   We divided global results into six regions (Africa, North and Central America, South America, South Asia, and South East Asia and Pacific). We further subdivided regions using biodiversity hotspots [47, 48] (S4 Appendix) and countries [49] (S5 Appendix).

**Bioclimatic potential for tropical dry forest.**   We used geospatial analysis tools to compile spatial data and develop binary raster maps (dry forest, non-dry forest) by applying the four bioclimatic definition of dry forest. Free and open-source Python software was used to manipulate raw data, primarily using Geospatial Data Abstraction Library and RasterIO. We used the cloud-based geospatial analysis platform Google Earth Engine [50] to analyze forest cover [27]. All analyses were compiled in a WGS84 projection and code used in our analysis is available in S6 Appendix.

**Statistical analyses.**   Regions, biodiversity hotspots, and country level data on area of tropical dry forest biome potential extent were examined for a normal distribution using one-sample Shapiro-Wilk normality test for small samples ($\leq$ 30) and Kolmogorov-Smirnov tests ($>$ 30). Non-parametric (Wilcoxon's rank sum test) tests were used to identify significant differences in area among bioclimatic definitions and between the WorldClim and CHELSA data sets for regions and biodiversity hotspots. Parametric (paired T-tests) were used to identify significant differences in area among bioclimatic definitions and between the WorldClim and CHELSA data sets for countries.

**Comparisons with WWF ecoregions, bioclimatic definitions, and field plots.** We calculated the area of each WWF ecoregion and compared results with four bioclimatic definitions (S1 Appendix). The agreement between WWF ecoregions, bioclimatic definitions, and field plots were measured using the percentage of dry forest plots that have been classified as dry forest according to the WWF ecoregions and different bioclimatic definitions (i.e. the percentage of true positives). However, we lack non-tropical dry forest plots such as tropical rain forests to calculate the percentage of false positives (i.e. non dry forest plots classified as dry forest) and true negative (i.e. non dry forest plots not classified as dry forest). Maximizing only the percentage of true positives might lead to selecting the best bioclimatic definition that predicts the larger extent of dry forest. Thus, we examine the percentage of true positives as a function of the extent of dry forest predicted by the different bioclimatic definitions. Given that the percentage of true positives likely depends on our sample of field plots, we used bootstrapping to compute a 95% confidence interval (CI) around the estimated values. Bootstrapping was done using 1,000 iterations and the lower and upper bounds of the CI were estimated using the quantiles 0.025 and 0.975, respectively.

**Tropical dry forest vegetation cover and change.** We selected the bioclimatic definition that had the highest agreement with field plots at a global spatial extent and calculated forest cover in 2000 and gross loss from 2001–2020 based on the global forest change data set using a forest cover threshold of $\geq 40\%$ to define closed canopy forest [17, 23, 27]. We also calculated similar results for (i) all tropical forests, (ii) the WWF tropical and subtropical dry broadleaf forest ecoregions, and (iii) the consensus (overlap) of the four bioclimatic definitions (aridity index, Murphy and Lugo, FAO, DryFlor). We also estimate the extent of forest cover using thresholds of $\geq 10\%$ [17] and $\geq 60\%$ [51, 52] for the bioclimatic definition that had the highest agreement with field plots at a global spatial for global comparisons (S3 Appendix).

## Results

### Comparisons of bioclimatic definitions of tropical dry forest biome

Estimates of tropical dry forest biome extent based on bioclimatic definitions varied (Table 2). The global extent estimated from the aridity index, Murphy and Lugo, and FAO climate

**Table 2. Estimate of tropical dry forest biome area (km²), globally and in different regions, based on four bioclimatic definitions (Murphy and Lugo, Food and Agriculture Organization of the United Nations [FAO], DryFlor, Aridity Index) using two projections.**

| Regions | Murphy & Lugo | FAO | DryFlor | Aridity Index |
|---|---|---|---|---|
| Global | 15,300,143 | 15,514,946 | 10,370,038 | 15,777,797 |
| | **16,123,939** | 15,177,193 | 10,820,627 | 14,376,146 |
| Africa | 6,825,248 | 7,480,815 | 5,005,193 | **8,000,948** |
| | 7,237,338 | 7,700,711 | 5,243,736 | 7,420,821 |
| North and Central America | 590,609 | 689,325 | 284,154 | **1,166,918** |
| | 652,516 | 811,289 | 326,525 | 1,028,800 |
| South America | 5,736,592 | 3,134,372 | 3,958,243 | 2,605,596 |
| | **6,042,593** | 3,774,250 | 4,023,236 | 1,866,315 |
| South Asia | 437,803 | 1,261,394 | 112,170 | **1,983,914** |
| | 481,186 | 1,230,178 | 153,111 | 1,417,872 |
| South East Asia/Pacific | 1,709,888 | 1,687,644 | 1,010,275 | 2,020,421 |
| | 1,710,303 | 1,660,763 | 1,074,017 | **2,642,338** |

WorldClim data presented above CHELSA. Highest extents in bold.

definitions were relatively similar (~15,000,000 km$^2$), while DryFlor covered the smallest area estimated ~10,000,000 km$^2$ (Table 2).

The aridity index using WorldClim and Murphy and Lugo using CHELSA estimated the largest extents of the tropical dry forest biome by region followed by the aridity index and FAO using CHELSA (Table 2). The boundaries of the tropical dry forest biome extent were more homogenous (smoother) for WorldClim than CHELSA data (Figs 2 and 3). Consensus maps showed that the area predicted to support tropical dry forest biome based on all four bioclimatic definitions (aridity index, Murphy and Lugo, FAO, DryFlor) was smaller than the combined non-consensus areas predicted by at least one bioclimatic definition. WorldClim contained larger areas of a consensus for all four bioclimatic definitions than CHELSA (Figs 2E and 3E).

On a regional scale, there was no significant difference in areas estimated by climatic data sets used (Wilcoxon's rank sum tests $p > 0.05$). Murphy and Lugo's definition estimated the largest tropical dry forest biome extent within biodiversity hotspots and countries (S4 and S5 Appendices). However, there was no significant difference between climatic data sets used in biodiversity hotspots (Wilcoxon's rank sum tests $p > 0.05$). At the country level, there was a significant difference in climatic data set used for all four bioclimatic definitions (paired t-test: Murphy and Lugo $p = 0.004$, FAO $p = 0.012$, DryFlor $p = 0.018$, aridity index $p = 0.005$).

## Comparisons to WWF ecoregions

The 53 WWF Ecoregions defined as tropical and subtropical dry broadleaf forest extend over an area of 2,918,256 km$^2$ with South Asia having the largest extent and Africa the smallest extent (Table 3). Globally, the aridity index and FAO definitions using WorldClim had the highest overlap with WWF Ecoregions (57%) followed by FAO using CHELSA (56%), and Murphy and Lugo using CHELSA (44%) (Table 3). DryFlor had the lowest overlap. There was high variation among regions, however. Murphy and Lugo and the aridity index had the highest overlaps within most WWF tropical and subtropical dry broadleaf forest ecoregions by region.

## Validation using field plots

We identified 540 field plots that met our search criteria. Most plots were between 0.03 and 0.1 ha, with diameter at breast height ranging from 1 cm to 10 cm. Species lists were available for 525 sites, however, full species density data were only available at 487 sites (S2 Appendix). Mean canopy cover estimates (1 km) were available for 534 sites (no forest cover data was available for the Mariana or Marquesas islands) and averaged 37.8% canopy cover, ± 25.9 in 2000 while maximum canopy cover estimates averaged 48.1 canopy cover, ± 30.1 (S2 Appendix).

Only 40% of the tropical dry forest field plots fell within the WWF tropical and subtropical broadleaf forest ecoregion boundaries (Table 4). This ranged from a low of 0% overlap in Africa to a high of 67% in South East Asia/Pacific. At a global scale, FAO bioclimatic definition using CHELSA had the highest agreement (81%) with field plots followed by Murphy and Lugo using CHELSA (75%). FAO (70%), Murphy and Lugo (66%) and the aridity index (64%) using WorldClim accounted for the next three highest overlaps with field plots. DryFlor had the lowest overlap (33–37%).

Most of the percentage of true positive were not correlated with the estimate of tropical dry forest biome extent (Fig 4). Thus, the better definition (e.g. the one which maximize the percentage of true positives) was not necessarily the definition that predicted the larger extent of the tropical dry forest biome. Globally the FAO-CHELSA was the best bioclimatic definition (80.9%), however, best definitions differ between the different regions. FAO-CHELSA was the

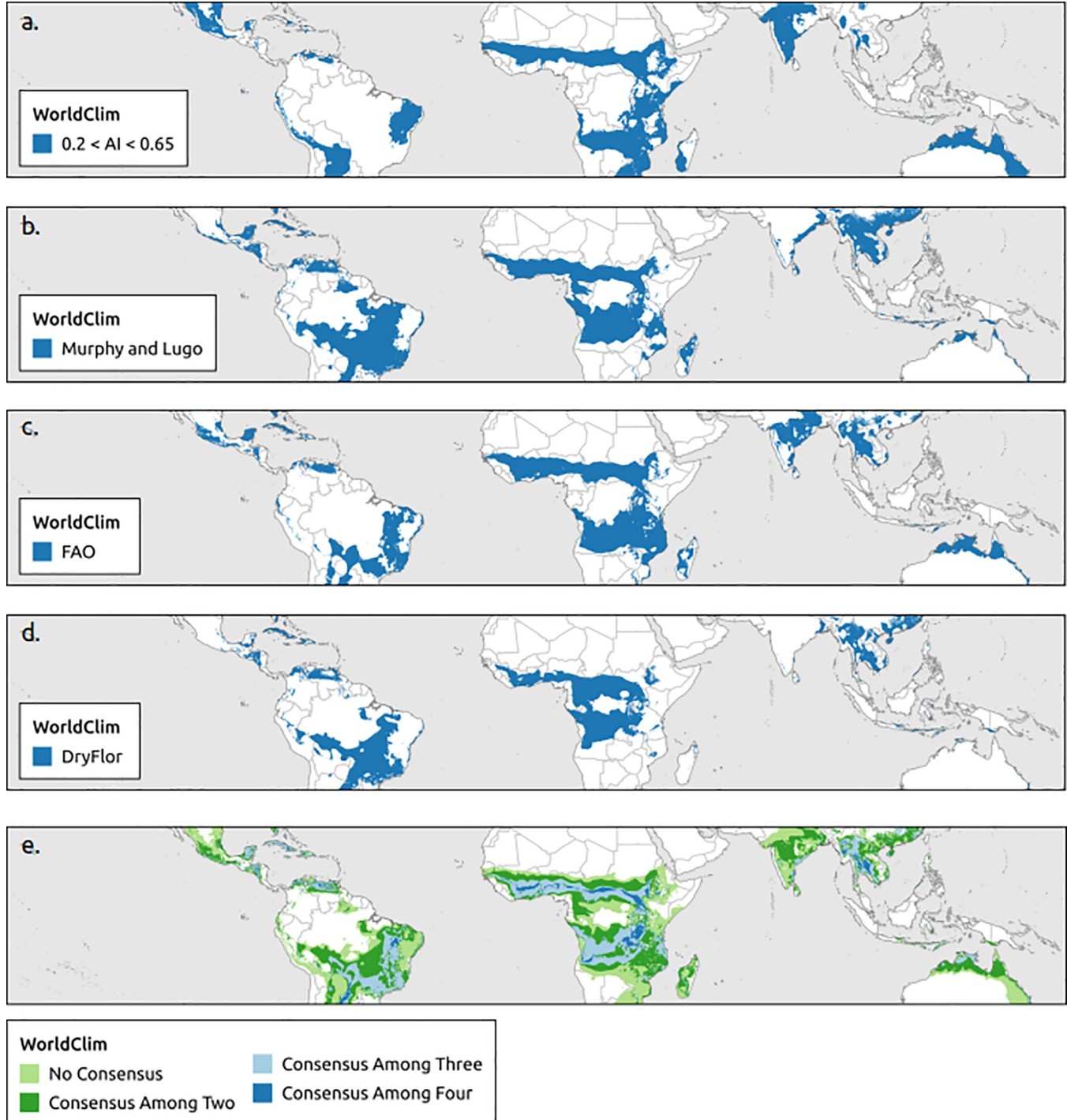

**Fig 2. Global distribution of tropical dry forest biome using WorldClim.** Based on (a) aridity index, (b) Murphy and Lugo, (c) Food and Agriculture Organization of the United Nations (FAO), (d) DryFlor bioclimatic definitions using WorldClim and (e) overlap of all four bioclimatic definitions (aridity index, Murphy and Lugo, FAO, and DryFlor).

best definition for North, Central (88.4%). There was no clear difference between FAO and Murphy and Lugo bioclimatic definitions in Africa, South America, and South Asia as evidenced by overlaps between confidence intervals. The percentage of true positives for

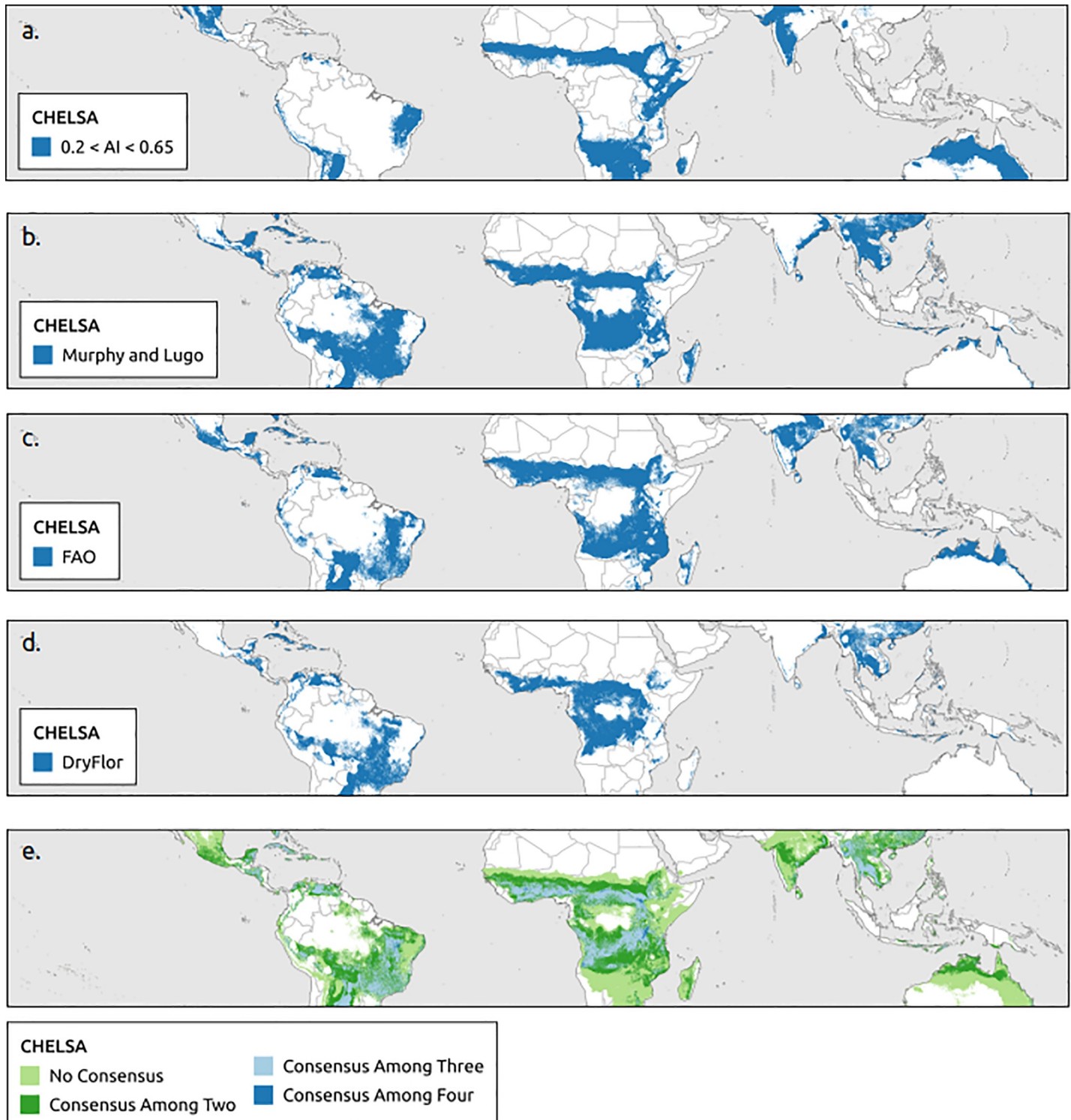

**Fig 3. Global distribution of tropical dry forest biome using CHELSA.** Based on (a) aridity index (b) Murphy and Lugo, (b) Food and Agriculture Organization of the United Nations (FAO), (c) DryFlor bioclimatic definitions using CHELSA and (e) overlap of all four bioclimatic definitions (aridity index, Murphy and Lugo, FAO, and DryFlor).

**Table 3. Extent of World Wildlife Fund's Tropical and Subtropical Dry Broadleaf Forest (TSDBF) ecoregion (km$^2$) and overlap (%) with bioclimatic definitions (Murphy and Lugo, Food and Agriculture Organization of the United Nations [FAO], DryFlor, Aridity Index).**

| Regions | TSDBF Ecoregion | Murphy & Lugo | FAO | DryFlor | Aridity Index |
|---|---|---|---|---|---|
| Global | 2,918,256 | 43% | **57%** | 22% | **57%** |
| | | 44% | 56% | 22% | 35% |
| Africa | 185,624 | **70%** | 69% | 3% | 48% |
| | | 60% | 56% | 1% | 13% |
| North & Central America | 511,057 | 45% | 63% | 23% | **67%** |
| | | 46% | 64% | 24% | 41% |
| South America | 642,243 | 58% | 47% | 31% | 43% |
| | | **60%** | 51% | 28% | 32% |
| South Asia | 983,944 | 9% | 60% | 2% | **82%** |
| | | 11% | 61% | 2% | 56% |
| South East Asia/Pacific | 593,608 | 75% | 53% | 49% | 24% |
| | | **76%** | 49% | 51% | 4% |

WorldClim data presented above CHELSA. Highest agreement with ecoregion in bold.

bioclimatic definitions were lower for South East Asia/Pacific where the TSDBF definition performed better (65.6%).

## Estimates of tropical dry forest cover and change

Using the FAO bioclimatic definition and CHELSA data set to estimate closed forest cover and change since 2000 (Fig 5), we estimated a total of 4,931,414 km$^2$ of closed canopy forest in 2000 and 4,369,695 km$^2$ of closed canopy forest in 2020. We estimate a potential gross loss of closed canopy tropical dry forest amounting to 561,719 km$^2$ (11%) between 2001 and 2020. Tropical dry forest was estimated to account for 25% of all closed canopy forest cover in the tropics in 2000 (Table 5), which is considerably larger than WWF tropical and subtropical dry broadleaf forest ecoregion estimates (782,492 km$^2$ or 4% of all tropical forests) and consensus maps based on all four definitions with either climatic dataset. Our estimates of tropical dry forest cover are greater than other global estimates that range from 1,048,700 km$^2$ of tropical dry forest ($\geq$ 40% closed canopy) [23] to 3,380,000 km$^2$ [17].

## Discussion

### WWF ecoregions

There was low overlap between WWF tropical and subtropical dry broadleaf forest ecoregions and bioclimatic definitions of tropical dry forest biomes (<57%) and field plots (40%) at a global scale. Tropical dry forest field plots occurred in other WWF ecoregions, including tropical and subtropical moist broadleaf forests (14% of the plots), tropical and subtropical coniferous forests (11%), tropical and subtropical grassland, savanna and shrubland (10%), and deserts and xeric shrublands (9%). This is of concern because the WWF ecoregions are widely used standard for delineating global biomes and estimating forest cover, canopy height, biomass, stand density, and anthropogenic disturbance [3, 23, 29, 31, 32, 53] (S1 Appendix).

We found that large areas of the tropical dry forest biome will be missed when using these WWF boundaries and that caution should be used when applying WWF boundaries for country scale analyses. At the country scale, higher resolution vegetation maps that identify forests, woodlands, shrublands, and savannas should be used. While WWF ecoregions may be appropriate for global or macro-scale analyses of tropical forests biomes, our results suggest that

**Table 4. Number of field plots and percent agreement with field plots (%) and World Wildlife Fund's Tropical and Subtropical Dry Broadleaf Forest (TSDBF) ecoregions and four bioclimatic definitions (Murphy and Lugo, Food and Agriculture Organization of the United Nations [FAO], DryFlor, Aridity Index).**

| Regions | Field Plots | TSDBF | Murphy & Lugo | FAO | Dryflor | Aridity Index |
|---|---|---|---|---|---|---|
| Global | 540 | 40% | 66% | 70% | 33% | 64% |
| | | | 75% | **81%** | 37% | 6% |
| Africa | 54 | 0% | 76% | **83%** | 52% | 41% |
| | | | 81% | 76% | 44% | 11% |
| North & Central America | 348 | 43% | 63% | 74% | 26% | 78% |
| | | | 77% | **89%** | 32% | 4% |
| South America | 48 | 33% | **79%** | 62% | 58% | 42% |
| | | | **79%** | 77% | 58% | 8% |
| South Asia | 35 | 57% | 91% | 91% | 34% | 77% |
| | | | 89% | **94%** | 31% | 14% |
| South East Asia/Pacific | 48 | 67% | **52%** | 31% | 42% | 8% |
| | | | 50% | 31% | 42% | 0% |
| **Biodiversity Hotspots** | | | | | | |
| Caribbean Islands | 227 | 49% | 62% | 73% | 28% | 87% |
| | | | 85% | **90%** | 37% | 1% |
| Cerrado | 17 | 0% | **100%** | 53% | 94% | 6% |
| | | | **100%** | 76% | 94% | 6% |
| Coastal Forests of East Africa | 12 | 0% | 83% | **92%** | 17% | 75% |
| | | | 67% | 75% | 25% | 8% |
| Guinean Forests of West Africa | 10 | 0% | **100%** | 80% | 90% | 0% |
| | | | **100%** | 80% | 50% | 0% |
| Indo Burma | 7 | 100% | **100%** | 43% | **100%** | 29% |
| | | | **100%** | 43% | **100%** | 0% |
| Madagascar & Indian Ocean | 1 | 0% | 0% | 0% | 0% | **100%** |
| | | | 0% | 0% | 0% | **100%** |
| Mesoamerica | 38 | 89% | 66% | **84%** | 16% | 82% |
| | | | 24% | 76% | 11% | 29% |
| New Caledonia | 5 | 100% | 80% | 40% | 80% | 0% |
| | | | 80% | **100%** | 80% | 0% |
| Polynesia/Micronesia | 26 | 62% | **42%** | 23% | 31% | 4% |
| | | | 35% | 15% | 27% | 0% |
| Tropical Andes | 5 | 20% | 40% | 40% | 40% | 40% |
| | | | 0% | **80%** | 20% | 40% |
| Wallacea | 3 | 100% | 67% | **100%** | 33% | 33% |
| | | | **100%** | 67% | 33% | 0% |
| Western Ghats & Sri Lanka | 10 | 0% | **100%** | 60% | **100%** | 10% |
| | | | **100%** | 90% | **100%** | 0 |

WorldClim data presented above CHELSA. Highest agreement with field plots in bold.

they may not be appropriate for analyzing tropical dry forest vegetation types or the impacts of climate change on tropical dry forests. Instead, the bioclimatic definitions of tropical dry forest from FAO with CHELSA datasets should be used.

## Bioclimatic definitions of tropical dry forest

According to our comparison with field plots data, the FAO and Murphy and Lugo bioclimatic definitions of tropical dry forest extent performed best. There were 358 field plots (66%) that

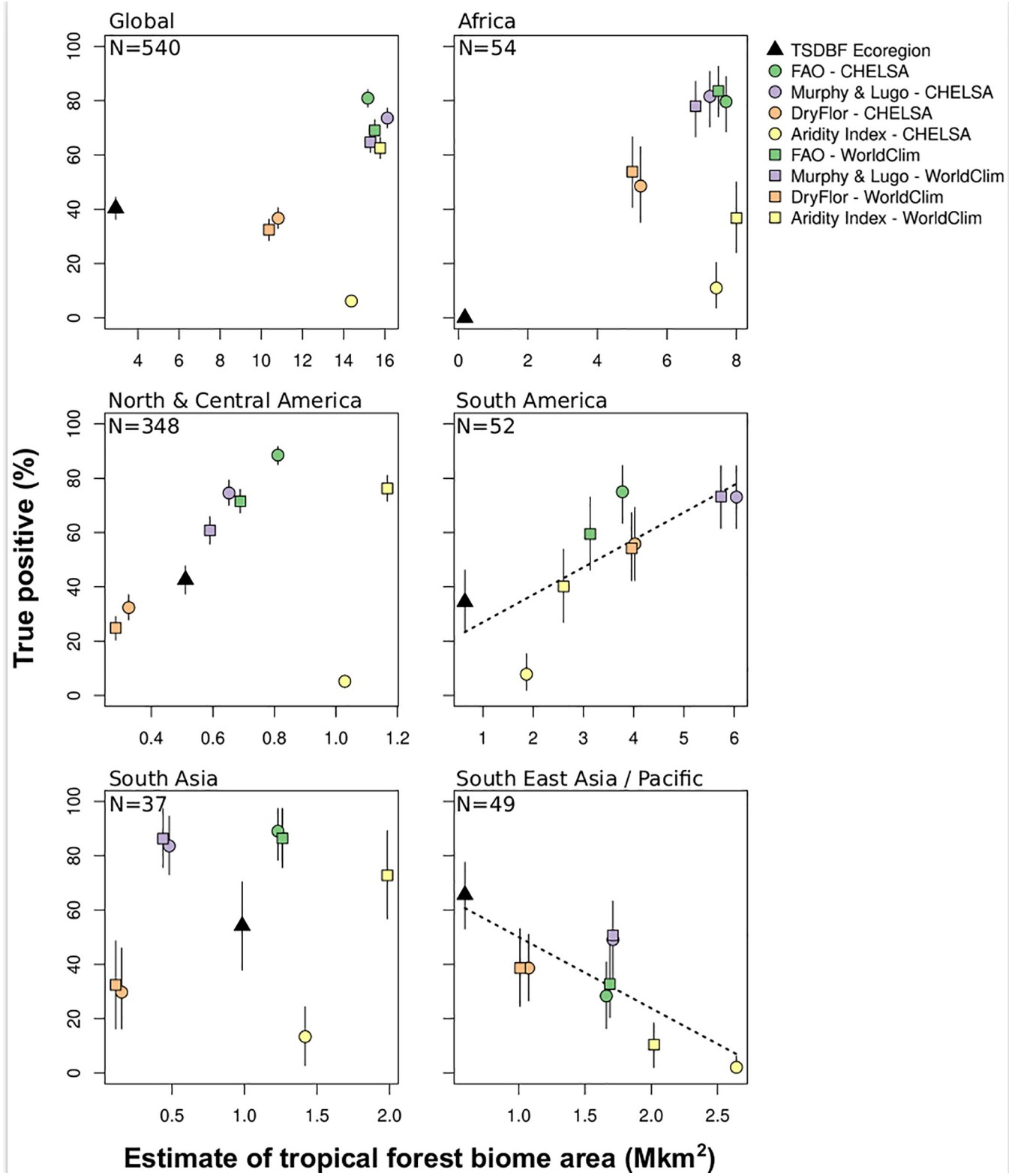

**Fig 4. Estimates of the percentage of true positives and biome extent using bootstrapping with 95% confidence intervals.** Points represent the average values and segments the lower and upper bounds of the intervals (i.e. the quantiles 0.025 and 0.975, respectively) using 1,000 iterations. Dotted black lines represent significant ($p < = 0.05$) linear relationships between the estimate of tropical dry forest biome area and the percentage of true positive.

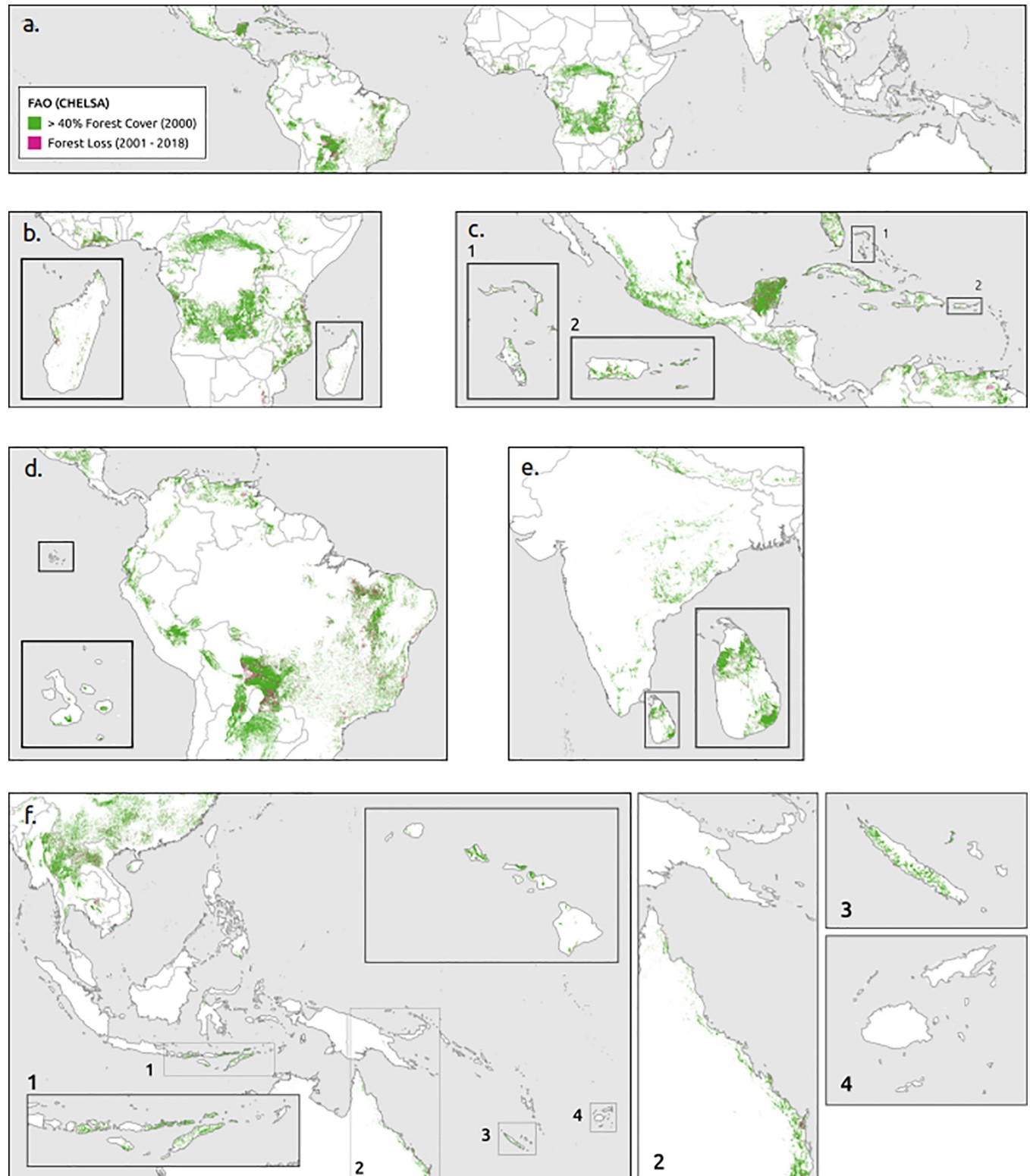

**Fig 5. Forest cover and change from FAO bioclimatic definition, CHELSA climate data set, and closed canopy cover.** (a) Global, (b) Africa, (c) North and Central America, (d) South America, (e) South Asia, and (f) Southeast Asia & Asia Pacific.

**Table 5. Comparisons of best methods for estimating closed canopy ($\geq$ 40% closed canopy) tropical dry forest extent and forest cover (km$^2$) in 2000 and 2020 between 30˚N and 30˚S.**

| Definition | Source | Forest Cover 2000 | Estimated Cover 2020 | Gross Loss 2001–2020 | Percent Gross Loss |
|---|---|---:|---:|---:|---:|
| Pantropics | Global Forest Cover | 20,064,659 | 17,962,101 | 2,102,558 | 10.48% |
| FAO | CHELSA | 4,931,414 | 4,369,695 | 561,719 | 11.39% |
| TSDBF | Wild Wildlife Fund | 782,492 | 670,336 | 112,156 | 14.33% |
| Consensus (all) | CHELSA | 20,758 | 19,166 | 1,592 | 7.67% |
| Consensus (all) | WorldClim | 332,336 | 305,973 | 26,363 | 7.93% |

FAO, Food and Agriculture Organization of the United Nations; CHELSA, Climatologies at High-resolution for the Earth's Land Surface Areas; TSDBF, tropical and subtropical dry broadleaf forest.

overlapped with both the FAO and Murphy and Lugo bioclimatic definitions (80 plots FAO only, 47 plots Murphy and Lugo only) and 54 field plots (10%), primarily in regions with long dry seasons (dry season > 9 months) or regions with over 2000 mm of mean annual precipitation such as Pacific islands, that did not overlap with either definition. The FAO definitions included drier regions with more seasonal forests while the Murphy and Lugo definition encompassed more moist regions with less seasonal forests. Thus, the FAO definitions appear to be best for identifying the core of tropical dry forest biome at a global spatial scale, while the Murphy and Lugo definition may perform best for identifying transitional areas between dry and moist or rain forest biomes. The DryFlor definitions worked well in the Neotropics but did not perform well at a global and regional spatial scales. The aridity index does not account for precipitation seasonality and appears to include the drier extremes of what would be considered tropical dry forest. Indeed, although forests that occur within an area with aridity index < 0.65 can be considered "tropical dry forests" because they occur in the tropics and clearly occur in dry conditions, the absence of a minimum precipitation threshold and seasonality criteria clearly misses a number of well-known tropical dry forest sites such as tropical dry forests of Central America and the Caribbean [2]. Future studies and comparisons of tropical dry forest biome should be undertaken using FAO bioclimatic definitions.

### Climatic data sets

WorldClim and CHELSA climate data sets did not significantly change estimates of tropical dry forest area based on all four bioclimatic definitions at the regional and biodiversity hotspot scales. However, using different climate data sets did identify different extents and areas at the country level (Figs 2 and 3). As analyses proceed to smaller spatial extents from regions to countries, the climatic data set used results in significantly different estimates of area. Thus the data set used becomes more important at small spatial extent. The CHELSA dataset generally had higher mean annual precipitation estimates than WorldClim but there was little difference in temperature estimates. The CHELSA data best matched field verified locations of tropical dry forest extent and offered some advantages over interpolated WorldClim data when assessing the future impacts of climate change. Since CHELSA is a downscaled global circulation models, it can easily be used to estimate the future climatic variables a 10 km and over last 4 million years at 10 km resolution [54]. The CHELSA data set also had the highest agreement with field plots and appears to be the most appropriate climate data set to use for studying tropical dry forests and possibly other tropical ecosystems.

### Tropical dry forest cover and changes

Our estimate of closed canopy ($\geq$ 40%) tropical dry forest in 2020 is 4.9 million km$^2$, based on the best performing parameters (FAO bioclimatic definition using CHELSA) and this is larger

than previous estimates. Miles et al. (2006) estimated 1,048,700 km$^2$ of tropical dry forest ($\geq$ 40% closed canopy) using MODIS imagery (500 m), while Bastin et al. (2017) estimated 3.38 million km$^2$ of closed canopy ($\geq$ 40%) dry forest in the tropics. Miles et al. (2006) estimate thus appears very conservative and more in line with our estimates in WWF ecoregions, but well above estimates of closed canopy forest based on consensus of all four climatic definitions. Although global comparisons of studies are difficult, it is clear that tropical dry forest vegetation types are experiencing a decline. Estimates based on all definitions of tropical dry forest (Table 5) are between 7% and 15% from 2000 to 2020 with closed canopy tropical dry forests experiencing high rates of deforestation and gross forest cover loss (S3 Appendix).

## Limitations

There is currently a lack of global comparative ground validation for our FAO CHELSA tropical dry forest biome and vegetation maps and field validation or very high resolution remote sensing validation is needed in the future [17]. While a growing number of tropical dry forest plots have been established worldwide, there is also a great deal of spatial bias in the number, extent, and density of tropical dry forest field plots. Our data set of 540 tropical dry forest plots were biased towards North and Central America, which accounted for 64% of the plots. Among the North and Central American plots, 65% were located in the Caribbean [8, 33]. There were relatively few comparative tropical dry forest plots from Madagascar (1 plot), eastern India, Indonesia, and Australia. Furthermore, there are a number of countries such as Angola, China, and Zambia that appear to have tropical dry forest biome and vegetation type but no field plots (S5 Appendix). This may be due to our search criteria for tropical dry forest plots which may have missed local names of dry forest vegetation types (e.g. monsoon forest), however, these areas deserve a high priority for establishing standardized plots in the future.

Almost all global climate data sets contain inaccuracies especially on islands with few weather stations and high heterogeneity in the landscape [15, 42]. Miranda et al. (2018) found that WorldClim data resulted in the misclassification of 15 to 20% of tropical dry forests in lowland South America while the addition of soil data improved classification by 3%. Delimiting the tropical dry forest biome is complex because tropical forest historically occurred across environmental and disturbance gradients, does not always have solid boundaries as depicted in GIS, and can be a continuum from drier to wetter areas. On the dry end of the spectrum, dry forests generally grade into savannas, shrublands, woodlands within the same climatic condition. The actual extent of vegetation within this region is highly impacted by substrate, soil moisture, and fire [19, 33]. On the wetter end of the spectrum, tropical dry forest generally transitions into moister forests, a zonal riparian forests, or swamps with increased canopy heights and an increasing number of tree species that are less susceptible to seasonal drought or dry soil conditions. These gradients clearly existed in the past but have been significantly impacted by humans certainly over the last 100 years [3, 4, 55]. It should be remembered that we are currently examining a disturbed landscape in most dry forest regions. Many dry forest regions are deforested, fragmented, and degraded based on estimate of forest cover over plots at a landscape scale (1 km) and it is difficult to precisely identify and delineate their distribution using global scale data (e.g. Hawaiian Islands).

Thus, climate definitions of tropical dry forest provide only a first order hypothesis and standard and repeatable method for identifying and estimating the spatial extent of tropical dry forest biome. Then analyses based on fine-scale features, such as forests (canopy height $\geq$ 3 m, $\geq$ 40% canopy cover), woodlands ($<$ 40% canopy cover), shrublands (stems $<$ 3 m), and savannas ($<$ 1 m, grasses) can be undertaken at a higher resolution such as 30 m from Landsat [27, 56] or very high resolutions $<$ 1 m in Google Earth [17, 50].

### Future research and applications

Our research focused on true positives (e.g. the percentage of dry forest plots that have been classified as tropical dry forest according to the different bioclimatic definitions) and false negatives (e.g. the percentage of dry forest plots that have not been classified as tropical dry forest according to the different bioclimatic definitions). However, future research at the biome level should include a confusion matrix with false positives (e.g. the percentage of non-dry forest plots, such as tropical rain forest, that have been classified as dry forest according to the different bioclimatic definitions), and the true negatives (e.g. the percentage of non-dry forest plots that have not been classified as dry forest according to the different bioclimatic definitions). This can be done at the biome level using high resolution locations of tropical rain forest plots.

Comparative analyses of biogeography, threats, and conservation can be undertaken using FAO (CHELSA) or Murphy and Lugo (CHELSA) boundaries as a baseline. It is well-established that biodiversity is greatly threatened by human activity [49, 57] and land cover changes such as those linked to human-induced forest loss, fragmentation, and degradation represent the largest current threat to biodiversity [57–59]. Miles et al. (2006) identified five global threat metrics for tropical dry forest including climate change (at a spatial resolution of 300 km), forest fragmentation (500 m), fire (10 km), agrosuitability (10 km), and population (10 km). Since this seminal work, there has been a significant increase in the temporal and spatial resolution of GIS and remote sensing data for these threat metrics, such as predicted future climate change (10 km), forest fragmentation ($\leq$ 30 m), fire and burned areas (375 m), agrosuitability and grazing (1 km), and population (1 km) [26, 60–64]. Thus, there are currently a number of global threat metrics that can be analyzed with FAO and CHELSA that might significantly improve our understanding of the health of tropical dry forests.

Twenty-five of the 36 global biodiversity hotspots appear to contain tropical dry forest biome based on three climatic definitions of tropical dry forests (S4 Appendix), possibly highlighting the importance of this biome and vegetation type. Analyses of protected areas, old growth forest, and threatened and endangered species are also needed within tropical dry forest biome and vegetation type. There has been a rapid increase in the number of protected areas and clearly there is a need to identify how well different regions are protected [23]. Using time series Landsat and fire data sets since 2000, it should be possible to identify stable closed canopy tropical dry forests that have not experienced fire or forest loss since 2000 [21, 63]. These forest areas may be some of the best preserved or relictual forests and contain an increasing rare combination of dry forest species. This may be especially true for regions like Africa, Asia, and Australia where fires and grazing are common and closed canopy dry forests are isolated in natural refugia [55].

### Conclusions

More precise knowledge of the extent and location of global tropical dry forest is needed to better understand forest cover dynamics, biodiversity threats and the conservation status of this vulnerable biome. We produced reliable data on tropical dry forest extent and overall forest loss spanning the last 20 years. We also demonstrate that nearly half of all tropical dry forest will be missed using the WWF ecoregion classification alone, with only 40% of our 540 tropical dry forest field plots falling within the WWF tropical and subtropical broadleaf forest ecoregion. We found that using the FAO bioclimatic definition applied to the CHELSA climate data set is a promising approach to identify potential dry forest extent based on the high agreement with field data. Closed canopy tropical dry forest currently occurs in 25 of the world's 36 biodiversity hotspots, a majority of pantropical countries, and has been experiencing high rates of deforestation (11%) between 2000 and 2020. Identifying the extent and distribution of tropical

dry forest regions are important for future research on the impacts of climate change and understanding the status of the world's tropical dry forest for conservation purposes.

## Supporting information

**S1 Appendix. Tropical and subtropical dry broadleaf forest defined by World Wildlife Fund.** Name of ecoregion, extent, and forest cover.
(CSV)

**S2 Appendix. Tropical dry forest field plots.** Comprehensive list of the 540 verified tropical dry forest plots, their coordinates, references, and bioclimatic characteristics.
(CSV)

**S3 Appendix. Forest cover and extent statistics.** Forest cover statistics for 540 tropical dry forest plots using resampled Global Forest Watch tree canopy cover (1 km$^2$) data for the year 2000 and a 1 km buffer around each plot. Total forest cover estimates and gross forest cover loss are also calculated for open canopy ($\geq$ 10%) and closed canopy ($\geq$ 40% and $\geq$ 60%) for FAO CHELSA from 2000 to 2020.
(DOCX)

**S4 Appendix. Biodiversity hotspots for mesoscale analyses of tropical dry forest.** Extent and agreement of tropical dry forest area.
(CSV)

**S5 Appendix. Tropical dry forest extent at country level.** Summaries of tropical dry forest extent based on climatic definitions and datasets at the country level.
(CSV)

**S6 Appendix. Computer code for global analysis of tropical dry forest.** Code used to accomplish a global analysis of tropical dry forest extent and cover based on a suite of remote sensing and modeled climate and forest data, as well as GIS layers of political and conservation boundaries.
(PDF)

## Acknowledgments

We gratefully acknowledge DryFlor.org (Karina Banda, Kyle Dexter, Danilo Neves, and Toby Pennington) for sharing advice and insights on data collection and neotropical dry forest inventories. We also would like to thank Kyle Cavanaugh for comments and suggestions.

## Author Contributions

**Conceptualization:** Jonathan Pando Ocón, Thomas Welch Gillespie.

**Data curation:** Jonathan Pando Ocón, Thomas Ibanez, Thomas Welch Gillespie.

**Formal analysis:** Jonathan Pando Ocón, Thomas Ibanez, Janet Franklin, Stephanie Pau, Gunnar Keppel, Gonzalo Rivas-Torres, Michael Edward Shin, Thomas Welch Gillespie.

**Funding acquisition:** Jonathan Pando Ocón.

**Investigation:** Jonathan Pando Ocón, Thomas Ibanez, Janet Franklin, Stephanie Pau, Gunnar Keppel, Gonzalo Rivas-Torres, Thomas Welch Gillespie.

**Methodology:** Jonathan Pando Ocón, Thomas Welch Gillespie.

**Project administration:** Jonathan Pando Ocón.

**Software:** Jonathan Pando Ocón.

**Supervision:** Jonathan Pando Ocón.

**Visualization:** Jonathan Pando Ocón, Thomas Ibanez, Gonzalo Rivas-Torres.

**Writing – original draft:** Jonathan Pando Ocón, Thomas Welch Gillespie.

**Writing – review & editing:** Jonathan Pando Ocón, Thomas Ibanez, Janet Franklin, Stephanie Pau, Gunnar Keppel, Thomas Welch Gillespie.

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
