## [Decision Letter · Decision Letter 0]

19 Mar 2021

PONE-D-20-37820

Global tropical dry forest cover and extent: A comparative study of bioclimatic definitions

PLOS ONE

Dear Dr. Ocón,

Thank you for submitting your manuscript to PLOS ONE. After careful consideration, we feel that it has merit but does not fully meet PLOS ONE’s publication criteria as it currently stands. Therefore, we invite you to submit a revised version of the manuscript that addresses the points raised during the review process.

ACADEMIC EDITOR: Please pay attention to the definition and range of tropical dry forests,and the methodsof data utilization.

We look forward to receiving your revised manuscript.

Kind regards,

RunGuo Zang

Academic Editor

PLOS ONE

Journal Requirements:

Reviewers' comments:

Reviewer's Responses to Questions

**Comments to the Author**

1. Is the manuscript technically sound, and do the data support the conclusions?

Reviewer #1: Partly

Reviewer #2: Yes

2. Has the statistical analysis been performed appropriately and rigorously? 

Reviewer #1: No

Reviewer #2: Yes

3. Have the authors made all data underlying the findings in their manuscript fully available?

Reviewer #1: Yes

Reviewer #2: Yes

4. Is the manuscript presented in an intelligible fashion and written in standard English?

Reviewer #1: Yes

Reviewer #2: Yes

5. Review Comments to the Author

Reviewer #1: Review of Ocon et al. Definition of Tropical Dry Forests

Overall comments:

There have been endless debates about what constitutes a tropical dry forest (especially as opposed to potentially similar vegetation such as savannas). The problem is partly compounded by the fact that tropical dry forests and savannas are highly dynamic and lie at the boundary of sharp vegetation transitions in some regions. The understorey may also change from predominantly grasses to woody plants within short time spans. Therefore, more objective criteria on defining tropical dry forests are certainly welcome. This study is a straightforward effort to try to delineate dry forests based on climatic criteria alone, which has its uses, particularly in large-scale analyses. I agree with the main conclusions of the study, namely, that the FAO definition of tropical dry forests seems to be the most appropriate. The much-cited Murphy and Lugo (1986) paper defines dry forests over an extreme range (250-2000 mm annually) of rainfall conditions.

However, I have several concerns regarding the methods.

Firstly, the authors do not account for the possibility that some of their validation plots may have been misclassified as dry forests. This is particularly true of some savanna regions (a lot has been written about this: see, for example, Ratnam et al., Proc. Roy. Soc. B, 2016). The authors should either account for this (for example, by checking sources for mention of % canopy cover and C4 grass cover) or discuss the limitation in detail. The authors must also discuss the possible circularity of some of the validation plot studies having adopted much-used definition such as Murphy & Lugo (1986) to classify their vegetation as dry forest. On a related note, the 40% canopy-cover cutoff to define a closed-canopy forest (L88, L219) needs a clearer justification or needs to be contrasted with results that use an alternate cutoff. Several studies treat < 60% as savanna and >= 60% as closed-canopy forest (e.g. Murphy & Bowman, Ecol. Lett., 2012).

Secondly, any study like this must assess results using a confusion matrix (proportions of false positives, true positives, false negatives, and true negatives). By only trying to maximize true positives and minimize false negatives, their criteria become overly liberal, potentially incorrectly including other vegetation types such as moist forest, savanna, shrublands, etc.

Finally, as they correctly note, the validation plot data set is highly biased in its geographical representation. While partly reflective of differences in research effort, it is also partly because their search criteria are inadequate as they miss several studies from South Asia and Indo-China (which host large extents of tropical dry forest) that should have been included.

Specific comments:

Line 52: should be “complemented” (not “complimented”)

Line 55-59: Seems out of place in the first paragraph of the Introduction. These do not provide any insight into the research questions but merely state the type of climate data sets available. I suggest these are shifted to the Methods section or at least reduced to a single sentence. In fact, I would go further and suggest that lines 72-108 are brought up front in the Introduction (after line 54) in order to clearly articulate the purpose of this study. The authors can then mention the availability of global data sets on climate, vegetation, land-cover and so on which could be used to resolve the questions.

Line 299: word missing in the sentence. In fact, there is a need for careful copyediting of the manuscript to correct minor errors.

Line 411-412: These categories are not mutually exclusive: C4 grasses can exist under a range of canopy covers.

Reviewer #2: The manuscript is an interesting address of the potential extent of the tropical dry forest biome based on bioclimatic definitions and climatic data sets to improve global estimates of distribution, cover, and change over time.

The data support the conclusions and answer the research question.

6. PLOS authors have the option to publish the peer review history of their article (what does this mean?). If published, this will include your full peer review and any attached files.

Reviewer #1: No

Reviewer #2: **Yes: **ADEL H YOUKHANA

---

## [Author Response · Author response to Decision Letter 0]

3 May 2021

Response to reviewers in blue

All authors would like to thank the reviewers for their time and comments and suggestions that significantly improve the manuscript. 

Reviewer #2

Review of the manuscript PONE-D-20-37820

The manuscript is an interesting address of the potential extent of the tropical dry forest biome based on bioclimatic definitions and climatic data sets to improve global estimates of distribution, cover, and change over time.

However, I have certain concern about:

-The authors try to do many comparisons in this manuscript but statistically there was no test used to do the real comparisons.

The reviewer is correct. The agreement between bioclimatic definitions and ground survey was measured using only the percentage of True Positives, i.e., the percentage of dry forest plots that have been classified as dry forest according to the different bioclimatic definitions. We also have the False Negatives, i.e., the percentage of dry forest plots that have not been classified as dry forest according to the different bioclimatic definitions. However, we don't have the False Positive, i.e., the percentage of non-dry forest plots that have been classified as dry forest according to the different bioclimatic definitions, and the True Negative, i.e., the percentage of non-dry forest plots that have not been classified as dry forest according to the different bioclimatic definitions.

So we have two solutions (1) we add other plots (tropical rain forests) to compute False Positives and True Negatives, (2) we keep focusing on True Positives and look at the True Positive/Extent ratios and we better acknowledge the limitation of our manuscript. 

Collecting high resolution location data for tropical rain forests at a global scale currently seems a bit outside the scope of this study. Thus we examine the True Positive/Extent ratios to identify if the largest extents of different bioclimatic definitions were associates with True Positives. We use a bootstrap procedure to calculate confident intervals around the percentage of True Positives and then test whether or not estimated percentage of True Positive is statistically greater than with the other bioclimatic definition.

We now include these statistics and the reviewer’s point in the Discussion section as limitations and future research. We are open to identifying tropical rain forest plots using a similar search criteria as our tropical dry forest plots (e.g. plot locations within 1 km) if absolutely needed. However, it does not seem possible to do a confusion metric for vegetation types such as savannas, woodlands, and shrublands plots because all three vegetation types can occur in the tropical dry forest biome. 

-Validation issue is important to be highlighted and there is lack of validation especially the ground validation.

This is an excellent point. The most important issue was the x, y of plot locations and we only selected plots for this study that had high resolution x, y data. However, there is a lack of ground validation for the FAO CHELSA biome and vegetation map and we now highlight this in the Discussion section. In theory we could use ICESAT or GEDI lidar data (covers 2% of the earth) to examine canopy heights to validate forests and compare results with FAO CHELSA forest cover maps or select a number of random points and undertake an assessment of forest cover and seasonality using very high resolution imagery in a similar fashion as Bastin et al 2017. However, this would take a bit more time. 

-The authors need to explain with more details, why they used WorldClim and CHELSA Climatologies.

The reviewer is correct that this needs to be highlighted up front in the Introduction. We used WorldClim because it the most widely used climatic dataset in biogeography. We used CHELSA because it is a downscaled global circulation model and a novel climatic dataset. However, CHELSA is based on global circulation models thus they are easier to mechanistically explain the results and patterns and use for modeling future climate change than WorldClim. We now highlight this in the Introduction, Methods, and Discussion. 

-More information about vegetation and plant cover is needed within study area.

Both reviewers raised this important point. We now provide data on estimates of percent canopy cover for all plots (mean, minimum, maximum forest cover at 1 km from Hansen’s 2000). This provides estimates of percent forest cover across the landscape. We also calculate > 10% and > 60% canopy cover for FAO CHELSA definition following Bastin et al. (2017) and Ratnam et al. (2016) as a comparison and provide a new AppendixS3 and further discussion. 

-Variability in selected plots with different sites and validation is problematic, the authors failed to consider and state the variability between the study sites, plots and locations, so the authors need to show that statistically (i.e, residual plots).

We now provide variability in the percent canopy cover over plots (x,y) at landscape scale of 1 km in 2000. This provides estimates of landscape variability over plots. 

-For the comparison within WWF Ecoregions and field plots, the authors need to show the accuracy level of the field plot location and within what range?

The accuracy of the WWF Ecoregions and field plots were assessed using the presence and absence of field plots within WWF Biome and Ecoregions polygons. We now include this information in Appendix S2 along with the WWF biome and ecoregion name. We are not sure how to estimate within what range and are open to ideas. Is this the distance from the TDF plots to the nearest WWF Tropical and Subtropical Broadleaf Forest? 

-Table 5. is showing the best methods for estimating closed canopy (>40% close), what about open canopy?

In this manuscript, we chose to mainly focus on closed canopy forests (> 40%) based on a standard definition following the FAO, Miles et al. 2006, and Bastin et al. 2017. Open canopy was not used as a criteria for tropical dry forest plot selection. However, we calculated open forest for the reviewer, update results to 2020, and include this as a new Appendix S3. 

-There is lack of information about tree species and density.

The reviewer is correct. We need to provide more information about tree species and density from the field plots. In the revisions of Appendix S2, we identify plot methods, area, and if they have standardize tree species and density data and include summaries in the Results section. 

- Abbreviations in the text are confusing and need to be simple.

The reviewer is correct. We have kept abbreviations in wide use in the literature (FAO, TDF, WWF, CHELSA, PET) and reduced abbreviations (e.g. GDAL) that are not in common usage. We also added the full names over each Table in the revised manuscript. Abbreviations used less than three times were removed. 

- The objectives are not clear and need to be more specific with some hypothesis and expectation for each objective.

The reviewer is correct. We now include specific expected results in the Problem Statement section. 

- Discussion part needs to be more specific related to the results.

The reviewer is correct. On re-examination, the Discussion paragraph order were 1) conclusion, 2) Tropical dry forest cover and changes, 3) WWF ecoregions, 4) Bioclimatic definitions of tropical dry forest, 5) Climatic data sets, 6) Limitations, 7) Future research and applications, and 8) conclusions again. We have reorganized the Discussion to follow the presented research objectives and Results and now it is organized as 1) WWF ecoregions, 2) Bioclimatic definitions of tropical dry forest, 3) Climatic data sets, 4) Tropical dry forest cover and changes, 5) Limitations, 6) Future research and applications, and 7) Conclusions.

Other Comments:

Title: Add “using two climatic data set” :

Global tropical dry forest cover and extent: A comparative study of bioclimatic definitions using two climatic data sets.

Thank you this. The title has now been changed to more accurately reflect the research undertaken. 

Introduction:

Line 63-65: need more details

Originally we stated. “Thus, climatic definitions allow delimiting the potential extent of biomes, but not detailed mapping of biome boundaries.” This was a mistake on our part. 

We now say “Thus, climatic definitions allow delimiting the potential extent of biomes, but not detailed mapping of vegetation boundaries”. 

Line 69-71: need citation.

We now provide two new citations. 

Hansen AJ, Neilson RP, Dale VH, Flather CH, Iverson LR, Currie DJ, Shafer S, Cook R, Bartlein PJ. Global change in forests: responses of species, communities, and biomes: interactions between climate change and land use are projected to cause large shifts in biodiversity. BioScience. 2001. Sept; 51(9):765-779.

Schmitt CB, Burgess ND, Coad L, Belokurov A, Besançon C, Boisrobert L, Campbell A, Fish L, Gliddon D, Humphries K, Kapos V. Global analysis of the protection status of the world’s forests. Biological Conservation 2009. Oct; 142(10):2122-2130.

Materials and Methods:

- Data collection needs more clarification

The reviewer is correct. We really spent a great deal of time identifying publications and datasets that contained 1) high resolution locations data, defined as TDF by authors, and contain floristic composition and structural data (canopy cover). We now expand on this in the Methods section and provide further data in the appendix on plot method used (FIA, Gentry), plot size, DBH and if there is species composition and density data. 

Reviewer #2: Review of Ocon et al. Definition of Tropical Dry Forests

Overall comments:

There have been endless debates about what constitutes a tropical dry forest (especially as opposed to potentially similar vegetation such as savannas). The problem is partly compounded by the fact that tropical dry forests and savannas are highly dynamic and lie at the boundary of sharp vegetation transitions in some regions. The understory may also change from predominantly grasses to woody plants within short time spans. Therefore, more objective criteria on defining tropical dry forests are certainly welcome. This study is a straightforward effort to try to delineate dry forests based on climatic criteria alone, which has its uses, particularly in large-scale analyses. I agree with the main conclusions of the study, namely, that the FAO definition of tropical dry forests seems to be the most appropriate. The much-cited Murphy and Lugo (1986) paper defines dry forests over an extreme range (250-2000 mm annually) of rainfall conditions.

However, I have several concerns regarding the methods.

Firstly, the authors do not account for the possibility that some of their validation plots may have been misclassified as dry forests. This is particularly true of some savanna regions (a lot has been written about this: see, for example, Ratnam et al., Proc. Roy. Soc. B, 2016). The authors should either account for this (for example, by checking sources for mention of % canopy cover and C4 grass cover) or discuss the limitation in detail. 

The reviewer is correct. There is very little comparative data on C4 grass cover for all TDF plot sites however all plots contain estimates of tree canopy cover (e.g. closed canopy). There are a number of way to calculate % canopy cover to insure plots were not misclassified as dry forests and compare results among sites. 

We now provide data on estimates of percent canopy cover over all plots (mean, minimum, maximum forest cover at 1 km from Hansen’s 2000). We also examined > 60 canopy cover for FAO CHELSA definition following Ratnam et al. (2016) as a comparison in an appendix (S3) and provide further discussion of the limitations. 

Ratnam J, Tomlinson KW, Rasquinha DN, Sankaran M. Savannahs of Asia: antiquity, biogeography, and an uncertain future. Philosophical Transactions of the Royal Society B: Biological Sciences. 2016. Sept 19; 371(1703):20150305.

The authors must also discuss the possible circularity of some of the validation plot studies having adopted much-used definition such as Murphy & Lugo (1986) to classify their vegetation as dry forest. 

This is an interesting point.  Indeed, the last author has collected a number of field plots in the Pacific loosely based on Murphy & Lugo (1986) climatic definition and an earlier version of the WorldClim data set.  However, all sites were selected based on local botanists identification of tropical dry forest.  However, we see your point.  Maybe other authors defined dry forest based on Murphy & Lugo (1986) definition in the past.  We now provide data on the number of plots that fell with both Murphy and Lugo and FAO definitions, only FAO, only Murphy and Lugo and plots that were outside both bioclimatic definitions.  There were 358 field plots (66%) that overlapped with both the FAO and Murphy and Lugo bioclimatic definitions (80 plots FAO only, 47 plots Murphy and Lugo only) and 54 field plots (10%), primarily in regions with long dry seasons (dry season > 9 months) or regions with over 2000 mm of mean annual precipitation such as Pacific islands, that did not overlap with either definition.  

On a related note, the 40% canopy-cover cutoff to define a closed-canopy forest (L88, L219) needs a clearer justification or needs to be contrasted with results that use an alternate cutoff. Several studies treat < 60% as savanna and >= 60% as closed-canopy forest (e.g. Murphy & Bowman, Ecol. Lett., 2012).

Both reviewers brought up this point. We used 40% canopy cover based on Miles et al. (2006) and Bastin et al. (2016) for comparison and we have now included this in the revisions. 

Miles L, Newton AC, DeFries RS, Ravilious C, May I, Blyth S, Kapos V, Gordon JE. A global overview of the conservation status of tropical dry forests. Journal of Biogeography. 2006 Mar;33(3):491-505.

Bastin JF, Berrahmouni N, Grainger A, Maniatis D, Mollicone D, Moore R, Patriarca C, Picard N, Sparrow B, Abraham EM, Aloui K. The extent of forest in dryland biomes. Science. 2017 May 12;356(6338):635-8.

We also include data on > 60 canopy cover for FAO CHELSA definition following Murphy & Bowman (2012) and Ratnam et al. (2016) as a comparison in a new Appendix and update the data on forest cover to 2020 in the revised manuscript.

Murphy BP, Bowman DM. What controls the distribution of tropical forest and savanna?. Ecology Letters. 2012. March; 15(7):748-758.

Ratnam J, Tomlinson KW, Rasquinha DN, Sankaran M. Savannahs of Asia: antiquity, biogeography, and an uncertain future. Philosophical Transactions of the Royal Society B: Biological Sciences. 2016. Sept 19; 371(1703):20150305.

Secondly, any study like this must assess results using a confusion matrix (proportions of false positives, true positives, false negatives, and true negatives). By only trying to maximize true positives and minimize false negatives, their criteria become overly liberal, potentially incorrectly including other vegetation types such as moist forest, savanna, shrublands, etc.

Both reviewers brought up this point. However, we do not have absence data for non-TDF plots (e.g. tropical rain forest). For instance, we do not have comparative data on high resolution locations of savannas or moist or mesic forests and this is especially true at a global spatial scale. We examined a number of global studies such as Slik et al. and Ibanez et al. but the resolution of their location data was generally 1 degree latitude which is very coarse. Furthermore, it is possible to have savanna, woodlands, and shrublands within 1 km pixel resolution of tropical dry forest. 

Thus we try to maximize true positives. We examine the True Positive/Extent ratios to identify if the largest extents of different bioclimatic definitions were associates with True Positives. We use a bootstrap procedure to calculate confident intervals around the percentage of True Positives and then test whether or not estimated percentage of True Positive is statistically greater than with the other bioclimatic definition.

We are not opposed to identifying about 500 globally distributed tropical rain forest plots with 1 km from Dryad or BIEN but this would take more time. 

Finally, as they correctly note, the validation plot data set is highly biased in its geographical representation. While partly reflective of differences in research effort, it is also partly because their search criteria are inadequate as they miss several studies from South Asia and Indo-China (which host large extents of tropical dry forest) that should have been included.

All methods for identifying published plots defined as “tropical dry forest” with high resolution x, y locations were undertaken in a standard and repeatable manner during the literature search. We did identify a number of vegetation surveys and plot data in South Asia and Indo-China but we could not use them because they lack x,y locations to within 1 km and many lacked closed canopy. Please let us know of any references we might of missed in South Asia and Indo-China. However, we now state that some plots in South Asia and Indo-China could be missed because they may be defined as ‘monsoon forest’. We now mention this in the discussion. 

Specific comments:

Line 52: should be “complemented” (not “complimented”)

This has been corrected in the revised manuscript. 

Line 55-59: Seems out of place in the first paragraph of the Introduction. These do not provide any insight into the research questions but merely state the type of climate data sets available. I suggest these are shifted to the Methods section or at least reduced to a single sentence. In fact, I would go further and suggest that lines 72-108 are brought up front in the Introduction (after line 54) in order to clearly articulate the purpose of this study. The authors can then mention the availability of global data sets on climate, vegetation, land-cover and so on which could be used to resolve the questions.

The reviewer is correct and we have reorganized the Introduction. Thank you. 

Line 299: word missing in the sentence. In fact, there is a need for careful copyediting of the manuscript to correct minor errors.

This has been correct in the revised manuscript along with a number of minor errors which we now highlight in Blue text. 

Line 411-412: These categories are not mutually exclusive: C4 grasses can exist under a range of canopy covers.

The reviewer is correct. We now identify grasses and structure as the structure (< 1 m) as the best way to identify the savanna vegetation type and provide a citation.

Alencar A, Shimbo JZ, Lenti F, Balzani Marques C, Zimbres B, Rosa M, Arruda V, Castro I, Fernandes Marcico Ribeiro JP, Varela V, Alencar I. Mapping three decades of changes in the brazilian savanna native vegetation using landsat data processed in the google earth engine platform. Remote Sensing 2020 March;12(6):924.

---

## [Editor Report · Decision Letter 1]

10 May 2021

Global tropical dry forest extent and cover: A comparative study of bioclimatic definitions using two climatic data sets

PONE-D-20-37820R1

Dear Dr. Ocón,

We’re pleased to inform you that your manuscript has been judged scientifically suitable for publication and will be formally accepted for publication once it meets all outstanding technical requirements.

Kind regards,

RunGuo Zang

Academic Editor

PLOS ONE
---

## [Editor Report · Acceptance letter]

12 May 2021

PONE-D-20-37820R1 

Global tropical dry forest extent and cover: A comparative study of bioclimatic definitions using two climatic data sets 

Dear Dr. Ocón:

I'm pleased to inform you that your manuscript has been deemed suitable for publication in PLOS ONE. Congratulations! Your manuscript is now with our production department. 

Kind regards, 

on behalf of

Professor RunGuo Zang 

Academic Editor

PLOS ONE